# Severe acute malnutrition among children under the age of 5 years

**Gift C. Chama**, **Lukundo Siame**, **Chanda Kapoma**, **Benson M. Hamooya**, **Sepiso K. Masenga** *

School of Medicine and Health Sciences, Mulungushi University, Livingstone, Zambia

* sepisomasenga@gmail.com

**Data Availability Statement:** All relevant data are within the paper and its Supporting information files.

## Abstract

### Background

Severe acute malnutrition (SAM) poses a significant threat to child health globally, particularly in low- and middle-income countries. Zambia, like many Sub-Saharan African nations, faces high rates of child malnutrition, with SAM contributing significantly to under-five mortality. Therefore, this study aimed to determine the prevalence and factors associated with SAM.

### Methods

This retrospective cross-sectional study was conducted at Livingstone University Teaching Hospital in Zambia (LUTH). SAM was defined according to the World Health Organization (WHO) criteria as either weight-for-height less than -3 standard deviations, mid-upper arm circumference (MUAC) less than 115 mm, or presence of bilateral pitting edema in children between 6 months and 5 years old who were attended to between 2020 and 2022. Data abstraction from pediatric patient records was conducted between August 2023 and January 2024. The records without the age and outcome variable were excluded. A total of 429 participants between 6 months and 5 years old were included, with demographic, clinical, and hematological parameters analyzed. Univariable and multivariable logistic regression were employed to investigate factors associated with SAM.

### Results

Overall, 429 medical records were included in the study and the prevalence of SAM was 27.0% (n = 116). Age group 6–24 months (Adjusted Odds Ratio [AOR]: 11.60; 95% Confidence Interval [CI]: 3.34–40.89, p<0.001), living with HIV (AOR:3.90; 95% CI: 1.14–13.70, p = 0.034), Tuberculosis (TB) (AOR:22.30, 95% CI: 4.53, 110.3, p < 0.001), comorbidities (AOR: 2.50; 95% CI 1.13, 5.88, p = 0.024) and platelet count (AOR: 1.00; 95% CI 1.00, 1.00, p = 0.027) were positively associated with SAM.

### Conclusions

This study found a high prevalence of SAM, exceeding the WHO target of reducing SAM to 5% by 2025. SAM was associated with younger age (6–24 months), HIV infection, TB,

**Funding:** The author(s) received no specific funding for this work.

**Competing interests:** The authors have declared that no competing interests exist.

comorbidities and platelet count. Therefore, there is need to enhance strategies aimed at reducing SAM among young children, children living with HIV, TB and comorbidities, particularly by intensive treatment, continuing and strengthening nutrition services.

## Background

Severe acute malnutrition (SAM) remains a critical global and national public health concern, particularly among children under the age of 5 years old [1]. Characterized by a significant deficit in weight-for-height ratio, bilateral pitting edema, or a mid-upper arm circumference (MUAC) below a certain threshold, SAM poses substantial risks to child health and development [2]. Globally, malnutrition affects millions of children, hindering their growth, cognitive development, and overall well-being [3, 4].

In low- and middle-income countries an estimated 34.2 million cases in 2022 of children under the age of five (5) suffered from SAM, and accounted for about 45% of fatalities in children under five years [5, 6]. Sub-Saharan Africa stands at the forefront of the malnutrition crisis, with a prevalence rate of SAM far exceeding global averages, especially among women and children [7]. Factors such as food insecurity, poverty, inadequate access to healthcare, and recurring environmental shocks exacerbate the vulnerability of children in this region [8]. Limited resources and infrastructure further impede effective interventions, perpetuating cycles of malnutrition and poverty across generations [9].

Zambia has high rates of child malnutrition and mortality in southern Africa with a prevalence estimated to be about 4.2% to 5% with SAM being a major contributor to under-five mortality, especially in rural parts of the country where it often goes undetected [10, 11]. Factors which lead to malnutrition are vast and include comorbidities such as HIV and tuberculosis (TB) infection, lack of food security, maternal nutritional practices, and lack of diet diversity [12]. In order to suggest strategies on how to fight malnutrition, there is a need to understand the factors associated with malnutrition [10]. Thus, this study aimed to determine the prevalence and factors associated with SAM among children under 5 years at Livingstone University Teaching Hospital (LUTH) pediatric department.

## Methods

### Study design and setting

This study was a retrospective cross-sectional study conducted at the pediatric department of LUTH. The department includes a malnutrition ward, a general ward, a High Dependency Unit (HDU), and a Neonatal Intensive Care Unit (NICU). LUTH serves as a referral center for southern and western provinces of Zambia, admitting around 1000 to 1500 children annually to the malnutrition and general ward.

### Eligibility and sampling method

Hospital records were abstracted from children between 6 months and less than 59 months. All children came through the outpatient department and once they were found to be malnourished or were in a serious state as determined by the attending doctor, they were hospitalized. By default, those with SAM were immediately hospitalized. The majority of children included in this study were therefore hospitalized. Variables of interest, including background demographic and clinical data were abstracted from patient's records. Any record with missing

data on age and outcome was excluded from the study. The records were chosen using a systematic sampling method where every second file from the patient's records in the department was selected for screening and eligibility. The chosen records were subsequently inputted into the research electronic data capture (REDCap).

## Sample size

We used openepi.com software to calculate the sample size. The minimal estimated sample size required was 414. We used an estimated prevalence of 5% of severe malnutrition in Zambia [13]. The confidence limits of 2.1% and design effect of 1. Total number of participants included were 429.

## Variables

SAM was defined according to WHO guidelines as the presence of either weight-for-height less than -3 standard deviations, mid-upper arm circumference (MUAC) less than 115 mm, or bilateral pitting edema in children between 6 months and 5 years old. This definition was used by attending clinicians for diagnosing SAM and admitting and treating children with the condition [14]. Moderate acute malnutrition (MAM) in children aged 6–59 months was diagnosed by either a weight-for-height between -3 and -2 Z-scores of the WHO Child Growth Standards median, or a mid-upper-arm circumference (MUAC) measurement between 115 mm and 125 mm by the attending clinician.

SAM was the dependent variable in this study. Various independent variables included socio-demographic factors (such as age, gender, and place of residence), medical conditions including HIV status and bacteriologically confirmed TB by smear microscopy, culture, GeneXpert, or lipoarabinomanannan assay, or clinical diagnosis (which included medical history, TB contact, physical examination, chest x-ray and a positive tuberculin skin test) when bacteriological criteria were not met [15], and comorbidities such as sickle cell disease (inherited disease characterized by the presence of hemoglobin S diagnosed by either solubility testing, peripheral smear, or hemoglobin(HB) electrophoresis [16]), acute watery diarrhea (defined as more than three loose or watery stools per day lasting less than 2 weeks [17]), anemia (defined as a hemoglobin level less than 11 g/dl [18]), pneumonia (diagnosis included presence of a cough or difficulty in breathing, plus at least one of the following: central cyanosis or oxygen saturation < 90% on pulse oximetry, severe respiratory distress, or signs of pneumonia with a general danger sign and evident chest radiographic findings such as presence of consolidation, infiltrates or effusion [19]), cerebral palsy, acute pharyngotonsillitis, acute kidney injury, congenital heart disease, asthma, diabetes mellitus and lymphoma. HIV and TB are the most common comorbidities in literature, so we excluded them from the list of common comorbidities and listed them separately. Additionally, the study examined parameters from a complete blood count analysis, comprising platelet count, neutrophil count, lymphocyte count, monocyte count, hemoglobin level, and white blood cell count.

Anthropometric measurement is crucial for monitoring a child's growth and involves assessing weight, length, height and body mass index [20]. To measure weight, a digital scale or analogy is used with the child undressed or minimally clothed on a calibrated scale [20]. Length (up to 2 years old) in infants and toddlers is measured with an infantometer, while height in older children (2 years and above) is measured using a stadiometer with a vertical bar [20]. BMI is calculated from weight and height. Measurements are plotted on age- and gender-specific growth charts to track growth patterns and identify any potential health concerns, using age-appropriate techniques to ensure the child's comfort during the process [20].

### Data collection

Data from patients' most recent hospital visit were collected between August 14, 2023 and January 8, 2024. Trained research assistants abstracted data from the medical records. The completeness of information was then audited for accuracy by senior data abstractors.

### Data analysis

Data for analysis were exported from the REDCap application to Microsoft Excel 2013 for data cleaning and then coded. Data analysis was performed using Stata version 15. Descriptive analysis was used to summarize categorical variables using frequencies and percentages. Continuous variables were summarized using median (interquartile range). Shapiro-wilk test was used to assess for data normality. Association of two categorical variables were determined using a chi-square and the Wilcoxon rank-sum test was used to determine the differences between two medians. Univariable and Multivariable logistic regression were used to investigate factors associated with SAM.

### Ethics

Mulungushi University School of Medicine and Health Sciences (SOHMS) Research Ethics Committee approved the study on 06[th] June 2023 (ethics reference number SMHS-MU2-2023-64) and LUTH administration gave permission to access patient's records. All data collected and analyzed were de-identified to ensure complete confidentiality. No information leading to identification of patients during and after analysis was abstracted and entered in the data collection form. Secondary data were used in this project thus, written/verbal consent was not applicable and was therefore waived by the ethics committee.

To strengthen the reporting for this observational study, we used the Strengthening the Reporting of Observational Studies in Epidemiology (STROBE) for reporting (S1 Checklist).

## Results

### Basic characteristic of the study participants

Out of an estimated 3000 files of children who were less than 15 years old that were attended to in the department between 2020 and 2021, we selected every second file and remained with 1500 files to screen for eligibility. The total number of files eligible and included in the analysis were 429, Fig 1.

Among the children 75.6% (n = 322) of the participants were aged between 6 to 24 months, Table 1. Of the participants, 55.9.8% (n = 240) were male, and the majority hailed from urban areas (n = 271, 63.2%). Among the 414 participants with a known HIV status, the prevalence of HIV was approximately 10.1% (n = 42). Forty-four (10.3%) participants were diagnosed with TB and the majority presented with pulmonary TB (n = 24, 54.5). Among the participants, the most common comorbidities observed were sickle cell disease (n = 13, 25%), followed by acute watery diarrhea (n = 12, 23.5.1%), anemia (n = 10, 19.6%), pneumonia (n = 9, 9.5%) and acute pharyngotonsillitis (n = 8, 8.4% %). Among the participants, 180 (26.3%) had malnutrition. Out of the cases, 116 (64.4%) participants were diagnosed with SAM, while 64 cases (35.6%) exhibited MAM. Majority (56%) of the children had weight for height Z score between greater or equal to or less than two to less than plus two. Hematological parameters revealed a median platelet count of 351 (IQR: 235, 450), neutrophil count of 5.0 (IQR: 3.0, 8.0), lymphocyte count of 4.0 (IQR: 2.6, 5.9), monocyte count of 0.8 (IQR: 0.5, 1.3), hemoglobin level of 10.3(IQR: 8.4, 11.4), and white blood cell count of 9.5 (IQR: 5.9, 12.5).

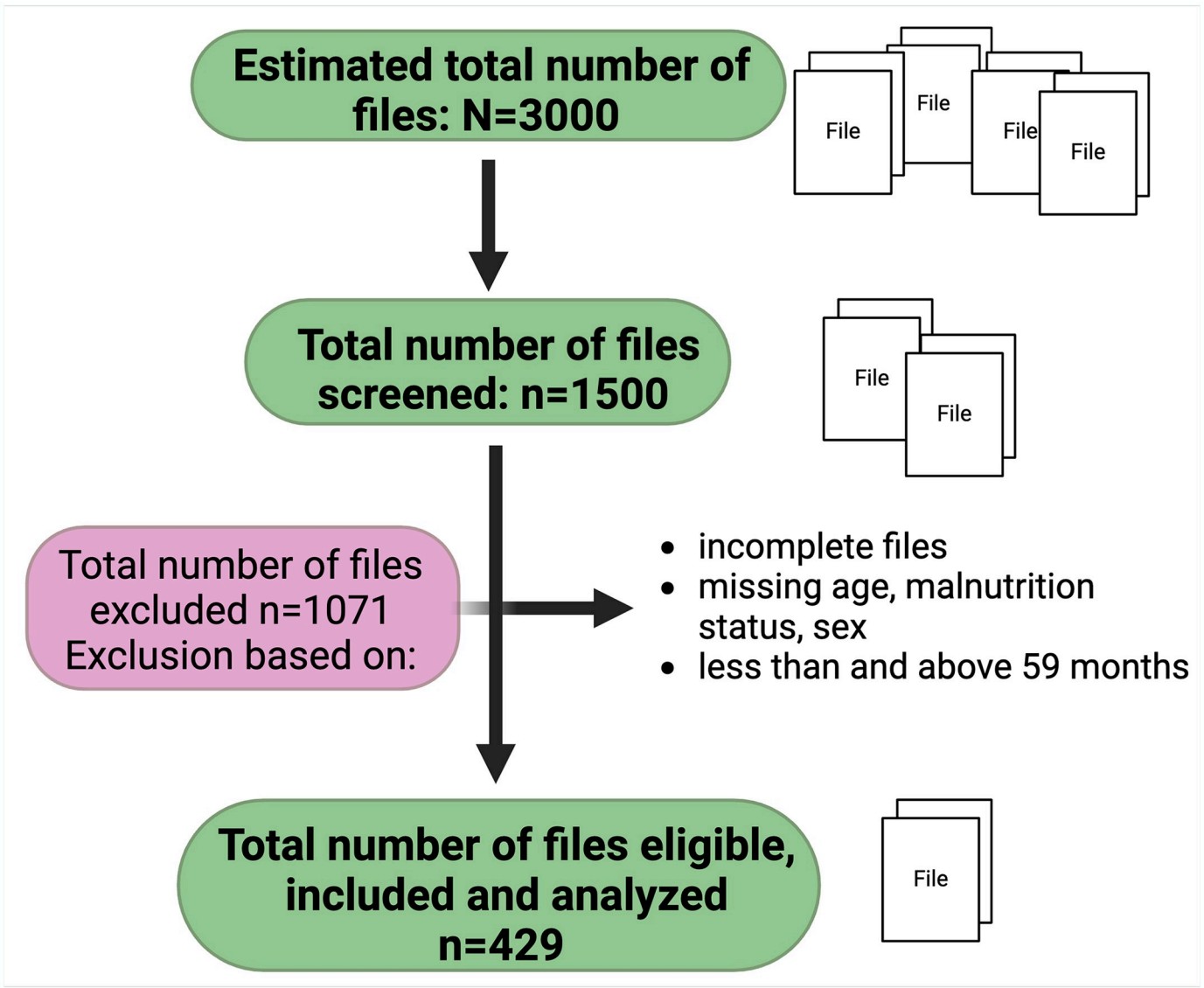

**Fig 1. Flowchart of screened and eligible files.**

### Relationship between SAM with sociodemographic and clinical characteristics of participants

The prevalence of SAM among the participants was 27.0% (n = 116), Table 1. As compared with participants without SAM, those with SAM were younger (median 16 vs. 19 months) and were living with HIV (54.8% vs. 23.1%). SAM was more prevalent among those with TB when compared to those without TB (68.2% vs. 21.8%). Additionally, having comorbidities was significantly associated with SAM when compared to those without SAM (77.7% vs. 40.2%).

The median white blood cell count was not significantly different between those with and without SAM (9.7 vs. 8.9 $x10^3$/µl, p = 0.154) Fig 2A. Neutrophil counts were slightly lower in patients with SAM compared to those without SAM (3.89 vs. 5.37 $x10^3$/µl, p = 0.039), Fig 2B. Conversely, Lymphocyte counts among patients with SAM were higher compared to those without SAM (4.06 vs. 3.72 $x10^3$/µl, p = 0.035), Fig 2C. The median monocyte count was not

Table 1. Sociodemographic and clinical characteristics of participants.

| Variable | Median, (IQR) OR frequency (%) | SAM | | P value |
|---|---|---|---|---|
| | | Yes = 116(27.0) | No = 313 (73.0%) | |
| **Age, months** | | | | |
| **6–24** | 322 (75.6) | 103 (32) | 219 (68.0) | **<0.001** |
| **25–59** | 107 (24.9) | 13 (12.1) | 94 (87.9) | |
| **Sex** | | | | |
| **Male** | 240 (55.9) | 59 (24.6) | 183 (75.4) | 0.293 |
| **Female** | 189 (44.1) | 55 (29.1) | 134 (70.9) | |
| **Residence** | | | | |
| **Urban** | 271 (63.2) | 67 (24.7) | 204 (84.8) | 0.256 |
| **Rural** | 158 (36.8) | 47 (18.9) | 111 (81.1) | |
| **HIV Status, *n = 414*** | | | | |
| **With HIV** | 42 (10.1) | 23 (54.8) | 19 (45.2) | **<0.001** |
| **Without HIV** | 372 (89.9) | 86 (23.1) | 286 (76.9) | |
| **Tuberculosis** | | | | |
| **Yes** | 44 (10.3) | 30 (68.2) | 14 (31.8) | **<0.001** |
| **No** | 385 (89.7) | 84 (21.8) | 301 (78.2) | |
| **Type of tuberculosis, *n = 44*** | | | | 0.287 |
| **Pulmonary** | 24 (54.5) | 18 (75.00) | 6 (25.0) | |
| **Extra pulmonary** | 20 (45.5) | 12 (60.0) | 8 (40.0) | |
| **Comorbidities** | | | | **<0.001** |
| **Yes** | 102 (23.9) | 41 (40.2) | 61 (59.2) | |
| **No** | 327 (76.2) | 73 (22.3) | 254 (77.7) | |
| **Top Comorbidities, *n = 52*** | | | | |
| **Sickle cell disease** | 13 (25.5) | 1 (7.7) | 12 (92.3) | 0.353 |
| **Anemia** | 10 (19.6) | 4 (40.0) | 6 (60.0) | |
| **Acute watery diarrhea** | 12 (23.5) | 3 (25.0) | 9 (75.0) | |
| **Pneumonia** | 9 (9.5) | 4 (44.4) | 5 (55.6) | |
| **acute pharyngotonsillitis** | 8 (8.4) | 2 (25.0) | 6 (75.0) | |
| **Malnutrition** | | | | |
| **Yes** | 180 (26.3) | | | |
| **No** | 249 (73.6) | | | |
| **Type of Malnutrition** | | | | |
| **SAM** | 116 (64.4) | | | |
| **MAM** | 64 (35.6) | | | |
| **Weight for Height Z score** | | | | |
| **> +3** | 5 (1.2) | | | |
| **≥+ 2 to <+3** | 8 (1.9) | | | |
| **≥ −2 to < +2** | 240 (56.3) | | | |
| **≥ −3 to <−2** | 84 (19.7) | | | |
| **<−3** | 89 (20.9) | | | |
| **Platelet count, *x10³ μl*** | 351 (235,450) | | | |
| **Neutrophil count, x10³ μl** | 5.0 (3.0, 8.0) | | | |
| **Lymphocyte count, x10³ μl** | 4.0 (2.6, 5.9) | | | |
| **Monocyte count, x10³ μl** | 0.8 (0.5, 1.3) | | | |
| **Hemoglobin, g/dl** | 10.3 (8.4, 11.4) | | | |

(*Continued*)

**Table 1.** (Continued)

| Variable | Median, (IQR) OR frequency (%) | SAM | | P value |
|---|---|---|---|---|
| | | Yes = 116(27.0) | No = 313 (73.0%) | |
| WBC, x10³ µl | 9.5 (5.9, 12.5) | | | |

Abbreviation: WBC White blood cells, SAM severe acute malnutrition, MAM Moderate acute malnutrition. Note: from a total of 44 cases of tuberculosis 30 cases were confirmed bacteriologically, while the rest of the cases were clinically diagnosed. While 89 with SAM were diagnosed using WHZ, 27 were clinically diagnosed.

significantly different between those with and without SAM (0.84 vs. 0.78 x10³/µl, p = 0.823), Fig 2D. Platelet levels were higher in individuals with SAM compared to those without SAM (394 vs. 311 x10³/µl, p = 0.006), Fig 2E. Hemoglobin levels (HB) were lower in individuals with SAM compared to those without SAM (9.0 vs. 10.6 g/dl, p = 0.002), Fig 2F.

## Univariable and multivariable analysis of factors associated with SAM

Table 2 shows results of univariable and adjusted multivariable analysis of factors associated with SAM. At univariable analysis, the age group 6–24 months were 3.4 times more likely to have SAM compared to the older group. Children living with HIV and TB had 4.4- and

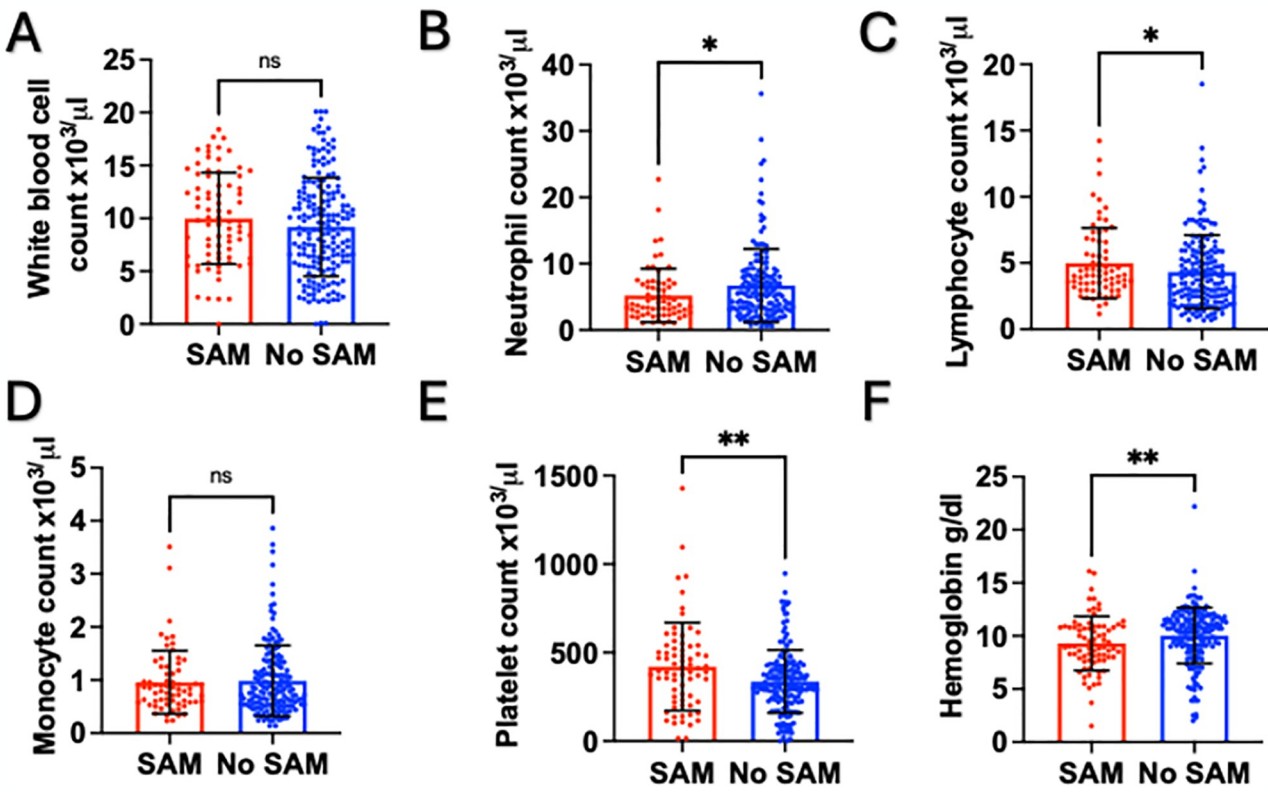

**Fig 2. Blood count parameters between those with and without severe acute malnutrition.** A. White blood cell count between those with (n = 77) and without SAM (n = 185). B. Neutrophil count between those with (n = 67) and without SAM (n = 167). C. Lymphocyte count between those with (n = 70) and without SAM (n = 170). D. Monocyte count between those with (n = 67) and without SAM (n = 167). E. Platelet count between those with (n = 71) and without SAM (n = 173). F. Hemoglobin level between those with (n = 78) and without SAM (n = 182). SAM, severe acute malnutrition. *p<0.05; **p<0.01; ns, p>0.05.

**Table 2. Univariable and multivariable logistic regression.**

|  | OR (95%Cl) | P-value | AOR (95%, Cl) | P-Value |
|---|---|---|---|---|
| **Age, months** |  |  |  |  |
| **6–24** | 3.40 (1.82, 6.36) |  | 11.60 (3.34, 40.89) | **<0.001** |
| 25–59 | 1 |  | 1 |  |
| **HIV Status** |  |  |  |  |
| Without HIV | 1 |  | 1 |  |
| With HIV | 4.40 (2.27, 8.42) | **<0.001** | 3.90 (1.14, 13.7) | **0.034** |
| **Tuberculosis** |  |  |  |  |
| No | 1 |  | 1 |  |
| Yes | 9.70 (4.71, 19.36) | **<0.001** | 22.3 (4.53,110.3) | **<0.001** |
| **Comorbidities** |  |  |  |  |
| No | 1 |  | 1 |  |
| Yes | 2.30 (1.40, 3.60) | **<0.001** | 2.50 (1.13, 5.88) | **0.024** |
| **Platelet count, x$10^3$ μl** | 1.00 (0.99, 1.00) | 0.180 | 1.00 (1.00, 1.00) * | **0.027** |
| **Neutrophil count, x$10^3$ μl** | 0.90 (0.87,1.00) | 0.050 | 0.90 (0.90, 1.25) | 0.162 |
| **Lymphocyte count, x$10^3$ μl** | 1.08 (0.99, 1.19) | 0.097 | 1.06 (0.23, 1.85) | 0.428 |

* rounded off to two decimal places. Original is 1.000 (1.000, 1.001)

9.7-times greater odds of having SAM compared to those without HIV and TB, respectively. Children with comorbidities were 2.3 times more likely to have SAM when compared to those without comorbidities.

At multivariable analysis the age group 6–24 months were 11.6 times more likely to have SAM when compared to those who were older. Children living with HIV had 3.9 times higher odds of having SAM compared to those without HIV. Children who were treated for TB had 22.3 times higher odds of having SAM compared to those without TB. Children with comorbidities were 2.5 times more likely to have SAM compared to those without. Every unit increase in platelet count was associated with increased odds of having SAM by 1.0 (AOR: 1.0: 95%Cl: 1.00, 1.0001 p<0.027).

## Discussion

The study objective aimed to determine the prevalence and correlates of SAM among pediatric patients at Livingstone University Teaching hospital. The present study found a prevalence of 27% of SAM. Other studies have reported varying prevalence rates of SAM; in Sudan, Mali and Ethiopia a prevalence of 6.5%, 4.4% and 21.2%, respectively, was reported [21–23]. Several factors likely contribute to the high burden of SAM in our setting, including a high prevalence of comorbidities like HIV and TB, limited access to healthcare, and low socioeconomic status further exacerbated by high food prices [13, 24–28].

In the present study, age was significantly associated with the nutritional status of the children. Children in the 6–24-month age group were more likely to have SAM. This finding is consistent with studies conducted in Nepal (2020), Nigeria (2020), and Ethiopia (2022) [26–28]. This may happen because younger children (6–24 months) have smaller stomachs compared to older children, requiring frequent meals to meet their energy demands [26, 27]. This period is also a transition phase between breastfeeding and complementary feeding, which in most cases is inadequate due to food unaffordability in our area [26, 29]. Additionally, this period is characterized by increased energy expenditure and nutritional demands [26, 27].

The present study revealed that children living with HIV were more likely to have SAM. This study is consistent with other studies which have shown similar results like in Burkina Faso (2017) and Kenya (2019) [30, 31]. HIV infection increases the body's nutritional needs by weakening the immune system, thus requiring more protein for immune cell production for metabolic processes [32]. Additionally, HIV and its treatments can impair nutrient absorption by damaging the intestinal lining, while chronic diarrhea caused by gut inflammation and damage further reduces nutrient absorption and contributes to fluid loss [32]. HIV also leads to muscle wasting by disrupting protein synthesis and metabolism [33]. This creates a vicious cycle, where malnutrition weakens the immune system, making individuals with HIV more susceptible to infections that worsen nutrient absorption and utilization [34].

TB has been shown to be associated with SAM as reported in Zambia (2017) and Nepal (2020) [35, 36]. Nutrition or its deficiency leads to impaired cell mediated and humoral immune responses, which in turn affects the ability of an individual to fight mycobacterium infection [37]. In our setting the incidence of TB is estimated at 333 cases per 100,000 population, ranking the country among the top 30 high-burden TB countries globally [38], hence the need to combat malnutrition.

Children with comorbidities were associated with SAM. The comorbidities prevalent were sickle cell disease, anemia, acute diarrheal disease, pneumonia and cerebral palsy consistent with studies by Baskaran *et al* (2018), Banga *et al* (2020) and Moate *et al* (2022) [39–41]. Studies have shown that severe malnourished children with comorbidities have a higher mortality rate than those without [42]. This increased risk is primarily due to immune dysfunction [43]. This dysfunction affects both innate and adaptive immunity. Innate immunity is compromised by a weakened skin, respiratory, and gastrointestinal mucosal barrier [44]. Additionally, alterations to gut microbiota occur [44]. Adaptive immunity is also affected, with reduced dendritic cells, blood complement factors, delayed type hypersensitivity, and low B-cells in the blood [44]. Further, there is increased apoptosis in lymphocytes, and lymphatic tissue shows atrophy [44]. Therefore, clinicians should prioritize close follow-up for these children, focusing on the critical comorbidities [42].

In the current study, a unit increase in platelet count was associated with SAM similar to a study conducted in Ethiopia (2020). The possible reason for thrombocytosis could be attributed to infections that likely increase levels of inflammatory cytokines which in turn heightened production of thrombocytes [45].

The study has certain limitations. It utilized a retrospective cross-sectional design conducted solely at Livingstone University Teaching Hospital, potentially limiting the generalizability of findings to LUTH. Secondly, the attending clinician's discretion, rather than the investigator's assessment, formed the basis for SAM classification. This approach can introduce bias, as attending physicians may underestimate or overestimate severity. Thirdly, MUAC is not taken routinely and therefore was not included in the diagnosis. Lastly, although the sample size was adequate for the study goal, a larger sample size would increase power and is recommended in future studies. However, despite these constraints, the study offers valuable insights into the burden and factors associated with SAM, serving as a foundation for future research endeavors.

## Conclusions

SAM remains high in our setting among under 5 years with younger age, HIV, TB comorbidities and platelet count positively associated. Therefore, targeted interventions are need to

reduce SAM among children living with HIV and TB, particularly by intensifying treatment, continuing and strengthening nutrition services.

## Supporting information

**S1 Checklist. Strobe checklist.**
(DOCX)

**S1 Dataset. Minimal dataset.**
(XLSX)

## Author Contributions

**Conceptualization:** Gift C. Chama, Lukundo Siame, Chanda Kapoma, Sepiso K. Masenga.

**Data curation:** Gift C. Chama, Lukundo Siame, Chanda Kapoma, Benson M. Hamooya, Sepiso K. Masenga.

**Formal analysis:** Gift C. Chama, Lukundo Siame, Chanda Kapoma, Benson M. Hamooya, Sepiso K. Masenga.

**Investigation:** Gift C. Chama, Sepiso K. Masenga.

**Resources:** Sepiso K. Masenga.

**Supervision:** Chanda Kapoma, Sepiso K. Masenga.

**Validation:** Gift C. Chama, Chanda Kapoma, Sepiso K. Masenga.

**Visualization:** Lukundo Siame, Chanda Kapoma, Benson M. Hamooya, Sepiso K. Masenga.

**Writing – original draft:** Gift C. Chama, Lukundo Siame, Chanda Kapoma, Benson M. Hamooya, Sepiso K. Masenga.

**Writing – review & editing:** Gift C. Chama, Lukundo Siame, Chanda Kapoma, Benson M. Hamooya, Sepiso K. Masenga.

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
