## [Decision Letter · Decision Letter 0]

3 Jun 2024

PONE-D-24-15995Prevalence and Factors associated with Severe Acute Malnutrition among children under the age of 15 years.PLOS ONE

Dear Dr.  Masenga,

Thank you for submitting your manuscript to PLOS ONE. After careful consideration, we feel that it has merit but does not fully meet PLOS ONE’s publication criteria as it currently stands. Therefore, we invite you to submit a revised version of the manuscript that addresses the points raised during the review process.

We look forward to receiving your revised manuscript.

Kind regards,

Stephen Michael Graham, FRACP, PhD

Academic Editor

PLOS ONE

Journal Requirements:

4. We note you have included a table to which you do not refer in the text of your manuscript. Please ensure that you refer to Table 1 and 2 in your text; if accepted, production will need this reference to link the reader to the Table.

Reviewers' comments:

Reviewer's Responses to Questions

**Comments to the Author**

1. Is the manuscript technically sound, and do the data support the conclusions?

Reviewer #1: Partly

Reviewer #2: Partly

2. Has the statistical analysis been performed appropriately and rigorously? 

Reviewer #1: Yes

Reviewer #2: Yes

3. Have the authors made all data underlying the findings in their manuscript fully available?

Reviewer #1: No

Reviewer #2: No

4. Is the manuscript presented in an intelligible fashion and written in standard English?

Reviewer #1: Yes

Reviewer #2: Yes

5. Review Comments to the Author

Reviewer #1: General comment

This paper describes the prevalence and identifies factors associated with severe acute malnutrition (SAM) among children <15 years in a regional hospital in Zambia. This is indeed a topic of public health importance given that severe malnutrition is the underlying cause of several childhood morbidities and contributes to mortality in Africa and other lower-resource settings.

The main problem of this paper is how the definition of SAM is applied in the study. Whereas the WHO definition of SAM applies to children between 6 months <5 years with specific criteria (presence of either WHZ <-3 standard deviation or MUAC<115mm or bilateral pitting oedema) the authors have applied only one of these parameters to define SAM, and have broadly used the definition to include children =>5 years with BMI for age <-3 SD who although technically are severely undernourished but they do not meet the definition of SAM.

I suggest the authors apply the definition as per WHO definition to children 6 months < 5 years and only include children in this age category in their analysis or use the term that can broadly encompass all forms of malnutrition or undernutrition if they wish to understand malnutrition more broadly in children <15 years.

Below are specific recommendations for each section:

Abstract

Background: Include a statement with the aim or hypothesis of the study.

Methods: Provide the definition of SAM and state inclusion criteria in this section

Results: Provide a summary of how many records were screened, and found eligible and clearly state the number of children the analysis is based on.

Introduction

Paragraph 3, Line 1: Review figures for the prevalence of undernutrition in Zambia which are much lower than those provided in the Zambia Demographic and Health Survey (https://www.zamstats.gov.zm/portfolio/zambia-demographic-and-health-survey-zdhs/) which provides a more countrywide representative figure than the reference used.

Methods

#Study design and setting:

A brief description of the hospital and the type of population served by the facility would help understand the settings in which this study was conducted.

#Eligibility/sampling:

Explicitly state what the study inclusion and exclusion criteria used. Were the participants selected hospitalised or recruited from ambulatory settings as well?

#Variables:

As stated in the general comments, the WHO criteria for SAM is: WHZ<-3 or MUAC<115 or the presence of bilateral pitting oedema in children 6mo <5years. Technically BMI for age <-3 in children above 5 years is described as “severe thinness” (https://www.who.int/tools/growth-reference-data-for-5to19-years/indicators/bmi-for-age )and is not interchangeable with SAM although it does provide a measure of severe undernutrition in this age group.

Were the classifications of SAM/malnutrition based on clinician diagnosis or the parameter (WHZ, BMI for age) calculated by the investigators using the clinical data? How were the data on the other parameters for SAM (MUAC or pitting oedema) dealt with?

# Sample size calculation:

• What was the rationale for inclusion of 10% non-response rate in a study that was primarily a retrospective review of patient records?

#On data collection:

• The procedures for data collection should be sufficiently described. How was the review of medical records structured? Was the SAM information extracted from the most recent medical visit or did it include any previous hospital visit? If there were multiple visits which instance was used?

• How was the screening of SAM structure in this institution? Do all children attending the hospital get a weight, height, and MUAC measurement?

• Did this review include hospital records over the stated study period only? If yes, how many <15 years were seen over this period in the hospital?

Results:

#Baseline characteristics

• Provide a summary of how many participant records were screened, found eligible, included, and analyzed. A flow chart could provide reason for exclusions of medical records.

• How many medical records were included in this analysis? Every 2nd record from a sample of 3000 implies there were 1,500 screened. How were the 695 finally selected for inclusion?

• Provide summary statements of the measures of malnutrition (WHZ, BMI) in the text or Table 1 to help understand the general nutritional status of the study cohort.

• The HIV prevalence among participants was approximately 9.2% (n=62). What was the denominator used here? 62/695 = 8.9%

• How many cases of tuberculosis were in this cohort? It’s not explicitly stated the number of tuberculosis cases in this cohort but only the proportion with pulmonary disease is provided in line 5 of the results.

• Both tables included are not referenced in the text in the results section to allow the reader to refer to the appropriate table.

• Table 1: Could you include how many TB cases were confirmed(bacteriologically) within the table or in the footnote?

Discussion:

• In paragraph 2, on the association between HIV and malnutrition, the authors reference adult studies only and do not refer to any relevant pediatric studies.

• In paragraph 6, apart from the retrospective nature of the study, where there any other limitations? How about the methods used to identify children with SAM? Some discussions on the limitations of the methods used to identify children with SAM should be included in this section.

• What is the implication of the findings for the care of children with malnutrition in this region or setting?

Reviewer #2: General comments:

The authors have chosen an important topic with high public importance and written the manuscript evaluating the prevalence and factors associated with SAM among children under the age of 15 years in an African country. The authors performed this study among under-15 children instead of under-5 children, although no rationality or data in favor of conducting this study in under-15 children are shown. The rationality should be elaboratively discussed in the background and briefly in abstract and need to have reflection all through the manuscript.

Specific comments:

Abstract

1. Please provide brief justification of conducting this study in under-15 children in the background

2. Please define SAM briefly in the methods

3. Usually abstract stands alone and thus before providing 16.7% prevalence in the results of abstract, the authors should provide the exact denominator and numerator here.

Background

4. Burden and fatality of SAM and related references used here are old and there are a number of recent data on these especially after 2022 and the authors should provide the most recent references all through the background.

5. The authors are requested to provide strong justification of conducting this study in under-15 children and need to have argument on how under-5 SAM aligns/differs to/from each other.

6. The background needs to have more connectivity with aims/objectives in the last paragraph.

Methods

Sample size

7. The evidence of 5% prevalence of SAM is used in calculating sample size in the Gambia, however, the prevalence was as high as 11% (https://www.lshtm.ac.uk/research/units/mrc-gambia/news/321426/mrc-nutrition-rehabilitation-centre-caring-children-severe-acute-malnutrition). The authors need to justify the selecting the lower prevalence for calculating the sample size.

Variables

8. There should have a brief description (a paragraph) on the procedure of the anthropometric measurement (weight, length/height, BMI) in children

9. Please correct acute diarrheal diarrhea as acute watery diarrhea with its definition

10. Please also define anemia, pneumonia, sickle cell disease, tuberculosis (bacteriological/clinical)

Data collection and data analysis

11. As data is a plural noun, please revise the grammar accordingly all through the text.

Results

Basic characteristic of the study participants

12. Definition of MAM should be provided in the methods before its use in the results.

13. Elaboration of SAM has already used earlier in the text, thus, this is redundant here.

14. All statistical values shown in the text of the results (Basic characteristic of the study participants and Univariable and multivariable analysis of factors associated severe acute malnutrition) are also available in the relevant tables. Please refer the tables for the statistical values rather duplicating these values in the text.

Discussion

15. Again, the elaboration of SAM has already used earlier in the text, thus, this is also redundant here.

16. The discussion section is less organized especially this section need to focus on most important observation with more explicit evidences.

17. The authors need to provide argument with evidences on how under-5 SAM aligns/differs to/from each other.

18. The observation of the association of an increase in neutrophil count with decreased odds of SAM should be discussed with more insights especially with the relation poor inflammatory responses in SAM.

18. Conclusion should be revised (deducting severe acute malnutrition or SAM) making a link with implication of the study.

6. PLOS authors have the option to publish the peer review history of their article (what does this mean?). If published, this will include your full peer review and any attached files.

Reviewer #1: No

Reviewer #2: No

---

## [Author Response · Author response to Decision Letter 0]

12 Jun 2024

Ref: RESPONSES TO REVIEWER’S COMMENTS

We would like to thank the reviewers for taking the time to make suggestions that have improved our manuscript. We have extensively revised the manuscript and addressed all concerns and suggestions. We now hope the current manuscript is acceptable for publication. Below are the point-by-point responses to all comments and suggestions.

Reviewer #1: General comment

This paper describes the prevalence and identifies factors associated with severe acute malnutrition (SAM) among children <15 years in a regional hospital in Zambia. This is indeed a topic of public health importance given that severe malnutrition is the underlying cause of several childhood morbidities and contributes to mortality in Africa and other lower-resource settings.

The main problem of this paper is how the definition of SAM is applied in the study. Whereas the WHO definition of SAM applies to children between 6 months <5 years with specific criteria (presence of either WHZ <-3 standard deviation or MUAC<115mm or bilateral pitting oedema) the authors have applied only one of these parameters to define SAM, and have broadly used the definition to include children =>5 years with BMI for age <-3 SD who although technically are severely undernourished but they do not meet the definition of SAM.

I suggest the authors apply the definition as per WHO definition to children 6 months < 5 years and only include children in this age category in their analysis or use the term that can broadly encompass all forms of malnutrition or undernutrition if they wish to understand malnutrition more broadly in children <15 years.

Response: Thank you, since most children presenting with SAM were under 5 years old, we have truncated the data to include only children between 6 months and under 5 years old, following the WHO SAM definition as suggested. We thank the reviewer for this suggestion.

Below are specific recommendations for each section:

Abstract

Background: Include a statement with the aim or hypothesis of the study.

Response: Thank you. We have done so. The aim of the study has been added 

Methods: Provide the definition of SAM and state inclusion criteria in this section

Response: we have now defined SAM and provided the inclusion criteria in this section.

Results: Provide a summary of how many records were screened, and found eligible and clearly state the number of children the analysis is based on.

Response: Thank you for this suggestion. We have now clarified this. In addition, we have adopted the reviewer’s earlier suggestion to truncate the data and include only include children 6months -<5 years

Introduction

Paragraph 3, Line 1: Review figures for the prevalence of undernutrition in Zambia which are much lower than those provided in the Zambia Demographic and Health Survey (https://www.zamstats.gov.zm/portfolio/zambia-demographic-and-health-survey-zdhs/) which provides a more countrywide representative figure than the reference used.

Response: Thanks very much: we have provided prevalence as per Zambia demographic survey published data. 

Methods

#Study design and setting:

A brief description of the hospital and the type of population served by the facility would help understand the settings in which this study was conducted.

Response: We have included this information. Thank you for the suggestion

#Eligibility/sampling:

Explicitly state what the study inclusion and exclusion criteria used. Were the participants selected hospitalized or recruited from ambulatory settings as well?

Response: Thank you for this suggestion. All children came through the outpatient department and once they were found to be malnourished or where in a serious state as determined by the attending doctor, they were hospitalized. By default, those with SAM were immediately hospitalized. The majority of children included in this study were therefore hospitalized. We have now clarified this under methods-eligibility criteria

#Variables:

As stated in the general comments, the WHO criteria for SAM is: WHZ<-3 or MUAC<115 or the presence of bilateral pitting oedema in children 6mo <5years. Technically BMI for age <-3 in children above 5 years is described as “severe thinness” (https://www.who.int/tools/growth-reference-data-for-5to19-years/indicators/bmi-for-age )and is not interchangeable with SAM although it does provide a measure of severe undernutrition in this age group.

Were the classifications of SAM/malnutrition based on clinician diagnosis or the parameter (WHZ, BMI for age) calculated by the investigators using the clinical data? How were the data on the other parameters for SAM (MUAC or pitting oedema) dealt with?

Response: Thank you for this suggestion. We have now clarified this in the manuscript. We applied the clinical and WHZ criteria to define SAM and we have now only included children 6mo <5years following WHO criteria. This was a good suggestion, and we thank the reviewer for pointing out this.

# Sample size calculation:

• What was the rationale for inclusion of 10% non-response rate in a study that was primarily a retrospective review of patient records?

Response: We apologize for using a terminology that would apply to primary data. What we meant by 10% non-response is Mainly to account for variables with incomplete/missing data especially on important variables such as the outcome. We have now corrected this.

#On data collection:

• The procedures for data collection should be sufficiently described. How was the review of medical records structured? Was the SAM information extracted from the most recent medical visit or did it include any previous hospital visit? If there were multiple visits which instance was used?

Response: thanks you for the suggestion. Data was collected from the most recent medical visits. We have now written it more clearly.

• How was the screening of SAM structure in this institution? Do all children attending the hospital get a weight, height, and MUAC measurement?

Response: thank you. All children attending the pediatric departments are measured for weight and height, however sometimes MUAC is not taken routinely and therefore not included. We have therefore included this in the limitation section 

• Did this review include hospital records over the stated study period only? If yes, how many <15 years were seen over this period in the hospital?

Response: No. The period stated was for data abstraction. The files were for patients in the year 2020 to 2022. We have now written out this more clearly in the manuscript. However, following your suggestion to truncate and only include <5 years, the total number of files reduced. We also want to clearly state that because of poor record keeping, and also that all files are kept together regardless of age (<5 years vs >5 years but <15years) the total number is an estimation from the records counted. Some files may have been missed at the time. We have included this as well in the limitation section. Thank you

Results:

#Baseline characteristics

• Provide a summary of how many participant records were screened, found eligible, included, and analyzed. A flow chart could provide reason for exclusions of medical records. How many medical records were included in this analysis? Every 2nd record from a sample of 3000 implies there were 1,500 screened. How the 695 were finally selected for inclusion?

Response: we have now included this information and a flowchart included as figure 1. This information is reported in the methods section and not results to avoid disrupting flow.

• Provide summary statements of the measures of malnutrition (WHZ, BMI) in the text or Table 1 to help understand the general nutritional status of the study cohort.

Response: thank you, the information has been added to table 1

• The HIV prevalence among participants was approximately 9.2% (n=62). What was the denominator used here? 62/695 = 8.9%

Response: thank you. The prevalence after reanalyzing was 10.1% (n=44) with determinator of 414. 15 records had missing variables on the HIV status. This is now indicated clearly

• How many cases of tuberculosis were in this cohort? It’s not explicitly stated the number of tuberculosis cases in this cohort but only the proportion with pulmonary disease is provided in line 5 of the results.

Response: Thank you, the information of number of TB cases have been included. 

• Both tables included are not referenced in the text in the results section to allow the reader to refer to the appropriate table.

Response: thank you. The tables have now been referenced

• Table 1: Could you include how many TB cases were confirmed (bacteriologically) within the table or in the footnote?

Response: thank you very much. The information has been added in the footnote of the table

Discussion:

• In paragraph 2, on the association between HIV and malnutrition, the authors reference adult studies only and do not refer to any relevant pediatric studies.

Response: thanks very. Appropriate reference pertaining to children have been added.

• In paragraph 6, apart from the retrospective nature of the study, where there any other limitations? How about the methods used to identify children with SAM? Some discussions on the limitations of the methods used to identify children with SAM should be included in this section.

Response: Thank you for the suggestion. Indeed that is limitation and has been included. 

• What is the implication of the findings for the care of children with malnutrition in this region or setting?

Response: targeted intervention are need to reduce SAM among children living with HIV and TB, particularly by intensifying treatment, continuing and strengthening nutrition services. Because its important, We have highlighted this in the conclusion. Thank you for the suggestion.

REVIEWER #2: GENERAL COMMENTS:

The authors have chosen an important topic with high public importance and written the manuscript evaluating the prevalence and factors associated with SAM among children under the age of 15 years in an African country. The authors performed this study among under-15 children instead of under-5 children, although no rationality or data in favor of conducting this study in under-15 children are shown. The rationality should be elaboratively discussed in the background and briefly in abstract and need to have reflection all through the manuscript.

Response: Thank you, since most children presenting with SAM were under 5 years old, we have truncated the results to include only children between 6 months and 5 years old, following the WHO SAM definition as suggested by reviewer 1.

Specific comments:

Abstract

1. Please provide brief justification of conducting this study in under-15 children in the background

Response: Thank you, since most children presenting with SAM were under 5 years old, we have now focused more on under 5 year patients as suggested by reviewer 1

2. Please define SAM briefly in the methods

Response: thank you, we have now defined SAM and provided the inclusion criteria in this section.

3. Usually abstract stands alone and thus before providing 16.7% prevalence in the results of abstract, the authors should provide the exact denominator and numerator here.

Response: thank you, we have added the information. 

Background

4. Burden and fatality of SAM and related references used here are old and there are a number of recent data on these especially after 2022 and the authors should provide the most recent references all through the background.

Response: thank you, the references have now been updated. 

5. The authors are requested to provide strong justification of conducting this study in under-15 children and need to have argument on how under-5 SAM aligns/differs to/from each other.

Response: Thank you, based on reviewer 1 suggestions this has now changed. since most children presenting with SAM were under 5 years old, we have now focused more on under 5-year patients as suggested by reviewer 1

6. The background needs to have more connectivity with aims/objectives in the last paragraph.

Response: we have revised accordingly and thank you for the suggestion.

Methods

Sample size

7. The evidence of 5% prevalence of SAM is used in calculating sample size in the Gambia, however, the prevalence was as high as 11% (https://www.lshtm.ac.uk/research/units/mrc-gambia/news/321426/mrc-nutrition-rehabilitation-centre-caring-children-severe-acute-malnutrition). The authors need to justify the selecting the lower prevalence for calculating the sample size.

Response: thank you, we calculated a using a prevalence of 5 % based on a study carried out in Zambia and based on country estimate of malnutrition by Zambia statistical agency of around 5%( https://www.zamstats.gov.zm/portfolio/zambia-demographic-and-health-survey-zdhs/) . 

Variables

8. There should have a brief description (a paragraph) on the procedure of the anthropometric measurement (weight, length/height, BMI) in children

Response: thank you for the suggestion. We have added this information. 

9. Please correct acute diarrheal diarrhea as acute watery diarrhea with its definition

Response: thank you. We have corrected the term accordingly and defined according to the guidelines 

10. Please also define anemia, pneumonia, sickle cell disease, tuberculosis (bacteriological/clinical)

Response: thank you. We have added the definition of these terms. 

Data collection and data analysis

11. As data is a plural noun, please revise the grammar accordingly all through the text.

Response: thank you, grammar has been revised accordingly. 

Results

Basic characteristic of the study participants

12. Definition of MAM should be provided in the methods before its use in the results.

Response: thank you, the definition has been added 

13. Elaboration of SAM has already used earlier in the text, thus, this is redundant here.

Response: thank you, correction has been made. 

14. All statistical values shown in the text of the results (Basic characteristic of the study participants and Univariable and multivariable analysis of factors associated severe acute malnutrition) are also available in the relevant tables. Please refer the tables for the statistical values rather duplicating these values in the text.

Response: thank you, correction has been made

Discussion

15. Again, the elaboration of SAM has already used earlier in the text, thus, this is also redundant here.

Response: thank you for the suggestion, correction has been made

16. The discussion section is less organized especially this section need to focus on most important observation with more explicit evidences.

Response: we have now only focused on significant results from multivariable analysis. Thank you

17. The authors need to provide argument with evidences on how under-5 SAM aligns/differs to/from each other.

Response: The study is now only around under 5 year olds as opposed to <15. We have also compared age groups 6- 24 vs 25 -59 for more meaningful or relevance.

18. The observation of the association of an increase in neutrophil count with decreased odds of SAM should be discussed with more insights especially with the relation poor inflammatory responses in SAM.

Response: thank you for the suggestion. The observation have been revised with more insight 

18. Conclusion should be revised (deducting severe acute malnutrition or SAM) making a link with implication of the study.

Response: thank very. The conclusion has been revised 

We have extensively revised the manuscript and addressed all concerns raised by the reviewers and editor. We want to thank you all again for the tremendous work and time that you committed in editing our work. Our manuscript is much improved, and we are very grateful.

All authors have reviewed and approved the manuscript.

Yours sincerely,

Prof Sepiso K. Masenga, BSc., MSc., PhD., FGHF., PGDip

---

## [Decision Letter · Decision Letter 1]

2 Jul 2024

PONE-D-24-15995R1Severe Acute Malnutrition among children under the age of 5 years.PLOS ONE

Dear Dr. Masenga,

Thank you for submitting your manuscript to PLOS ONE. After careful consideration, we feel that it has merit but does not fully meet PLOS ONE’s publication criteria as it currently stands. Therefore, we invite you to submit a revised version of the manuscript that addresses the points raised during the review process.

Kind regards,

Steve

Stephen Michael Graham, FRACP, PhD

Academic Editor

PLOS ONE

Journal Requirements:

Reviewers' comments:

Reviewer's Responses to Questions

**Comments to the Author**

1. If the authors have adequately addressed your comments raised in a previous round of review and you feel that this manuscript is now acceptable for publication, you may indicate that here to bypass the “Comments to the Author” section, enter your conflict of interest statement in the “Confidential to Editor” section, and submit your "Accept" recommendation.

Reviewer #1: (No Response)

Reviewer #2: (No Response)

2. Is the manuscript technically sound, and do the data support the conclusions?

Reviewer #1: Yes

Reviewer #2: Yes

3. Has the statistical analysis been performed appropriately and rigorously? 

Reviewer #1: Yes

Reviewer #2: Yes

4. Have the authors made all data underlying the findings in their manuscript fully available?

Reviewer #1: No

Reviewer #2: No

5. Is the manuscript presented in an intelligible fashion and written in standard English?

Reviewer #1: Yes

Reviewer #2: Yes

6. Review Comments to the Author

Reviewer #1: The authors have addressed the issues raised in the first review> A few points for clarification and consideration below.

Thank you for considering the suggestion on the inclusion of the participants' flow chart. According to the strobe guidance flow charts describing the participants should be included in the results sections

Table 1: Include the means of the haematological parameters comparing those with SAM vs No SAM

Could the authors review the confidence intervals for the adjusted analysis for the platelet count in table 2?

Reviewer #2: The overall quality of the revised manuscript improved a lot. The authors should be congratulated for that. However, I have minor concerns about the following three definitions that need to be revised before the acceptance of this manuscript.

1. The authors mentioned that clinical diagnosis of TB was made when bacteriological criteria were not met but the authors did not mention the clinical criteria used for clinical TB diagnosis. These criteria should be precisely written.

2. The authors defined acute watery diarrhea as ‘frequent loose or watery stools lasting less than 2 weeks’ which is not precise as this should be defined as ‘more than three loose or watery stools per day lasting less than 2 weeks (WHO Pocket Book 2013, 2nd edition, page 125)’

3. The authors defined pneumonia as ‘a lung infection caused by bacteria or viruses, diagnosed using either a chest radiograph or clinically through vital sign measurement’, this is not a precise definition at all. The authors should use the better reference (PMID: 15976876) for chest radiograph defining pneumonia and must need to mention the clinical features used for diagnosing pneumonia/severe pneumonia. The authors may take the help from WHO Pocket Book 2013, 2nd edition, page 80).

7. PLOS authors have the option to publish the peer review history of their article (what does this mean?). If published, this will include your full peer review and any attached files.

Reviewer #1: No

Reviewer #2: No

---

## [Author Response · Author response to Decision Letter 1]

2 Jul 2024

To the reviewers and Editor,

Ref: RESPONSES TO REVIEWER’S COMMENTS

We would like to thank the reviewers for taking the time to make suggestions that have improved our manuscript. We have now made revisions to the minor comments in the manuscript and incorporated all suggestions. We now hope the current manuscript is acceptable for publication. Below are the point-by-point responses to all comments and suggestions.

Reviewer #1: General comment

Thank you for considering the suggestion on the inclusion of the participants' flow chart. According to the strobe guidance flow charts describing the participants should be included in the results sections

Response: We have now moved the flow chart to the results section. Thank you for this suggestion

Table 1: Include the means of the haematological parameters comparing those with SAM vs No SAM

Response: We have included them. Thank you. 

Could the authors review the confidence intervals for the adjusted analysis for the platelet count in table 2?

Response: The confidence interval are 1.000 (1.000, 1.001) so we just rounded them off to 2 decimal places to maintain consistence with other values in the table. For clarity, we have included this in the footnote. Thank you.

Reviewer #2: The overall quality of the revised manuscript improved a lot. The authors should be congratulated for that. However, I have minor concerns about the following three definitions that need to be revised before the acceptance of this manuscript.

1. The authors mentioned that clinical diagnosis of TB was made when bacteriological criteria were not met but the authors did not mention the clinical criteria used for clinical TB diagnosis. These criteria should be precisely written.

Response: Thank you for your kind remarks. We have now written the clinical diagnosis. The criterion included a medical history, TB contact, physical examination, chest x-ray and a positive tuberculin skin test). Thank you!

2. The authors defined acute watery diarrhea as ‘frequent loose or watery stools lasting less than 2 weeks’ which is not precise as this should be defined as ‘more than three loose or watery stools per day lasting less than 2 weeks (WHO Pocket Book 2013, 2nd edition, page 125)’

Response: Thank you for the precise definition. We have now included it.

3. The authors defined pneumonia as ‘a lung infection caused by bacteria or viruses, diagnosed using either a chest radiograph or clinically through vital sign measurement’, this is not a precise definition at all. The authors should use the better reference (PMID: 15976876) for chest radiograph defining pneumonia and must need to mention the clinical features used for diagnosing pneumonia/severe pneumonia. The authors may take the help from WHO Pocket Book 2013, 2nd edition, page 80).

Response: Thank you for suggesting the precise definitions. We have now included. Much appreciated.

We have revised the manuscript and addressed all concerns raised by the reviewers. We want to thank you all again for the tremendous work and time that you committed in editing our work. Our manuscript is much improved, and we are very grateful.

Yours sincerely,

Sepiso K. Masenga, BSc., MSc., PhD., FGHF., PGDip

Associate Professor of Human Pathology,

Chair/Director of Hypertension, HIV/AIDS, Nutrition, Diabetes and Dyslipidemia (HAND) Research Group

Coordinator Postgraduate programs

Head of Physiological Sciences Department

Academic Editor – Frontiers in Cardiovascular Medicine

Editorial board member - PLOS ONE

Editorial board member – BMC infectious diseases

---

## [Decision Letter · Decision Letter 2]

7 Aug 2024

Severe Acute Malnutrition among children under the age of 5 years.

PONE-D-24-15995R2

Dear Dr. Masenga,

We’re pleased to inform you that your manuscript has been judged scientifically suitable for publication and will be formally accepted for publication once it meets all outstanding technical requirements.

Kind regards,

James Mockridge

Staff Editor

PLOS ONE

Reviewers' comments:

Reviewer's Responses to Questions

**Comments to the Author**

1. If the authors have adequately addressed your comments raised in a previous round of review and you feel that this manuscript is now acceptable for publication, you may indicate that here to bypass the “Comments to the Author” section, enter your conflict of interest statement in the “Confidential to Editor” section, and submit your "Accept" recommendation.

Reviewer #2: All comments have been addressed

2. Is the manuscript technically sound, and do the data support the conclusions?

Reviewer #2: Yes

3. Has the statistical analysis been performed appropriately and rigorously? 

Reviewer #2: Yes

4. Have the authors made all data underlying the findings in their manuscript fully available?

Reviewer #2: No

5. Is the manuscript presented in an intelligible fashion and written in standard English?

Reviewer #2: Yes

6. Review Comments to the Author

Reviewer #2: The authors kindly addressed all the issues raised by the reviewers. Thus, the revised manuscript may be accepted in its present form.

7. PLOS authors have the option to publish the peer review history of their article (what does this mean?). If published, this will include your full peer review and any attached files.

Reviewer #2: No

---

## [Editor Report · Acceptance letter]

14 Aug 2024

PONE-D-24-15995R2 

PLOS ONE

Dear Dr. Masenga, 

I'm pleased to inform you that your manuscript has been deemed suitable for publication in PLOS ONE. Congratulations! Your manuscript is now being handed over to our production team.

Kind regards, 

on behalf of

Dr James Mockridge 

Staff Editor

PLOS ONE